# Self-Reported Purchasing Behaviour, Sociodemographic Predictors of Plant-Based Protein Purchasing and Knowledge about Protein in Scotland and England

**DOI:** 10.3390/nu14214706

**Published:** 2022-11-07

**Authors:** Magdalena M. E. Brandner, Claire L. Fyfe, Graham W. Horgan, Alexandra M. Johnstone

**Affiliations:** 1The Rowett Institute, University of Aberdeen, Aberdeen AB25 2ZD, UK; 2Biomathematics and Statistics Scotland, Aberdeen AB25 2ZD, UK

**Keywords:** plant-based foods, purchasing behaviour, plant-based protein, sociodemographic factors, legumes, sustainable eating, whole-foods approach, protein, consumer behaviour

## Abstract

Plant-based diets are seen as a food-based strategy to address both the impact of dietary patterns on the environment, to reduce climate change impact, and also to reduce rates of diet-related disease. This study investigated self-reported consumer purchasing behaviour of plant-based alternative foods (PBAF) and wholefood plant protein foods (legumes) with a cross-sectional online survey. We identified the sociodemographic factors associated with purchasing behaviour and examined knowledge about protein and plant-based diets. We recruited and obtained consent from *n* = 1177 adults aged >18 from England and Scotland (mean age (± standard deviation (SD)) 44 (16.4) years), across different areas of social deprivation, based on postcode. Descriptive statistics were conducted, and sociodemographic factors were examined by computing covariate-adjusted models with binary logistic regression analysis. A total of 47.4% (*n* = 561) consumers purchased PBAF and 88.2% (*n* = 1038) wholefood plant-proteins. The most frequently purchased PBAF were plant-based burgers, sausages, and mince/meatballs. Individuals from low deprivation areas were significantly more likely than individuals from high deprivation areas to purchase wholefood plant-proteins (odds ratio (OR) 3.46, *p* = 0.001). People from low deprivation areas were also more likely to recognise lentils as good source of protein (OR 1.94, *p* = 0.003) and more likely to recognise plant-based diets as healthy (OR 1.79, *p* = 0.004) than those from high deprived areas. These results support current trends of increasing popularity of PBAF, which is positive for the environment, but also highlights these products as being ultra-processed, which may negatively impact on health. The study also re-enforces the link between deprivation, reduced purchasing of wholefood plant-proteins and knowledge of plant-based protein and diets. Further research is needed to examine healthfulness of PBAF and how sociodemographic factors, especially deprivation, affect both food choice and consumption of wholefood plant-proteins.

## 1. Introduction

Global action is required to reduce greenhouse gas (GHG) emissions to limit the damage of global warming. Both the climate and environment are severely endangered by our current food systems and dietary approaches, particularly in the developed world [1]. The production and consumption of meat, and other animal-derived foods and beverages, significantly contribute to GHG emissions, both in the UK and worldwide [1,2]. At the same time, societies face high rates of diet-related disease due to consumption of excess calories from foods high in energy, also salt, saturated fats and added sugar. These are often foods that are highly processed and account for 25–60% of daily energy intake in many countries [3]. In the UK, a suboptimal diet is estimated to cause a loss of 1.5 million years of healthy lives [4] and one in seven deaths, contributing to worldwide 11 million deaths related to poor dietary habits [5,6]. Affordability, accessibility and availability of unprocessed plant-based protein-rich foods remain a challenge to support healthy and environmentally sustainable diets [7]. Plant-based diets are viewed as an important food strategy by science and policy makers to tackle both health and environmental problems and achieve healthy, sustainable diets and a resilient food system [8,9]. Plant-based diets are widely seen as diets that emphasise a variety of plant foods and simultaneously attempt to reduce or even exclude animal-derived foods [10]. Plant protein sources can be wholefood plant proteins, like legumes (peas, beans, lentil or soya), wholegrains, nuts and seeds [11] or plant-based alternative foods (PBAF). PBAF are made from plant-proteins and are defined, “to mimic the taste and texture of their animal-based counterpart” [12]. Example products are meat-free burgers or sausages, Quorn^TM^ (Stokesley, UK) products or milk alternatives. This analysis specifically focuses on the purchasing of wholefood plant proteins in the form of legumes and PBAF (Quorn^TM^, soya products, other meat-free products). Quorn^TM^ is a brand of mycoprotein meat alternatives (Stokesley, UK), referred to as mycoprotein for the remainder of the paper.

There is a scarcity of research on recent consumer behaviour relating to plant-protein choices. A recent review from Alae-Carew et al. in 2022 provides insight, by analysing the consumption of PBAF [12] in the UK, reporting females, millennials and those with higher income as having significantly higher PBAF intake. Despite many systematic reviews having explored attitudes and acceptance around the consumption of more sustainable protein-foods or reduced meat intake [13,14,15], it is often highlighted that studies neglect comparisons of a multitude of sociodemographic variables beyond gender or cultural background [15]. These factors are important to consider to ensure a healthy, safe and equitable food system for all [4,16]. This poses as an opportunity for our current research to consider other sociodemographic variables such as country of residence, gender, age, ethnic background, but also deprivation levels. Socioeconomic status (SES) or index of deprivation (IMD) have been repeatedly identified in the literature as important factors driving food choice and behaviour [17,18]. More evidence is needed regarding plant-based eating, in order to support consumers facing deprivation, to encourage a transition to both healthy and sustainable eating. Finally, it was also identified that there is a paucity of research concerning consumer knowledge about protein, its physiology and health-benefits [19]. Taking the above into consideration, more information is needed about consumer behaviour, sociodemographic predictors of purchasing behaviour, as well as consumer’s knowledge of protein. This knowledge could inform public health strategies and messages to enable transition towards purchasing and consuming more plant-based foods [7]. Therefore, this study aims to (1) contrast consumer purchasing trends of plant-based protein products in Scotland and England (2) analyse sociodemographic factors within consumer behaviour of plant-based protein purchasing, i.e., gender, age, ethnic background and IMD and (3) explore consumers’ knowledge about protein, i.e., sources, physiological role and health benefits.

## 2. Materials and Methods

### 2.1. Data Source and Sample

Data for this study originated from the cross-sectional survey, to assess UK attitudes, beliefs and trends in plant-based choices, from the study, “What Plants are on your plate” conducted in April 2022 on the Qualtrics market research platform. Data aimed to gather responses from 1000 adults (aged > 18 years), fluent in English, as *n* = 500 from England and Scotland. Participants were members of the Qualtrics Research Panel. This is a pre-recruited sample of panelists similar to the population of a census [20]. This ensured random participants of an adult population in England or Scotland participated in the survey. The study questionnaire is included in Appendix A. 

### 2.2. Sociodemographic Data

Sociodemographic information including country of residence, age, gender, IMD were collected as part of this online questionnaire. Country of residence could either be selected as Scotland or England and gender was categorised during analysis into male, female and other. IMD levels were derived from postcode data and coded into quintiles with 1 being the least deprived and 5 being the most deprived. In England and Scotland IMD levels provide a relative measure of deprivation derived from information of several aspects of deprivation, such as income, employment, education, health deprivation and disability and even barriers to housing and services, crime and living environment [21]. Age was collected as self-reported numerical data and was combined into generation groups, which were, Generation Z (age 18–23), Millennials (age 24–39), Generation X (age 40–55), Baby Boomers (age 56–74) and Traditionalists (75+ years), similar to those described by Alae-Carew et al. in adults [12].

### 2.3. Purchasing Data and Protein Knowledge Data

Consumer purchasing behaviour data was obtained in the survey using multiple-choice lists of wholefood plant-proteins and processed plant-based alternative food products to indicate whether these are purchased in general grocery shops. Further specification data of kind and state (e.g., canned, frozen, dried) was also chosen (see questionnaire in Appendix A). 

Categorical answers to statements regarding protein’s physiological role and health benefits were also included in this study, showing agreement, neutrality or disagreement with protein sources, environmental and health prevention benefits of protein. 

### 2.4. Data Analysis

We used descriptive statistics to describe purchasing behaviour and characteristics (using IBM Corporation, released 2021, version 28.0, IBM SPSS Statistics for Windows, Armonk, NY, USA). Participants were categorised into two grouping factors: PBAF purchasers and wholefood protein purchasers. They were categorised as PBAF purchasers if they indicated purchasing any alternative plant-protein products (mycoprotein products, soya products, other meat free products). Similarly, participants were categorised as wholefood protein purchasers if they indicated purchasing any legume products (green beans or peas, beans or lentils). To compare these two purchasing groups, chi-square tests for trend and continuity correction were conducted.

Factors affecting the grouping (PBAF purchasers and wholefood plant-protein purchasers) were tested using binary logistic regression models to determine associations between the consumption and sociodemographic covariates. Initially, they were analysed in separate univariate models for each sociodemographic factor. Following this a multivariate analysis in one model, adjusted for country of residence, gender, age, ethnicity and IMD, was carried out. Furthermore, the number of purchased product types from each PBAF purchasing and wholefood plant-protein were scored and distributions across sociodemographic groups were analysed using non-parametric Mann–Whitney and Kruskal–Wallis tests. Throughout all tests, statistical significance was set at *p* < 0.05. 

For the third research objective, data were described using descriptive statistics, and non-parametric Mann–Whitney and Kruskal–Wallis tests were carried out to compare between different sociodemographic groups and their agreement with the protein knowledge statements. In addition, for comparability of sociodemographic covariates with both knowledge statements and purchasing behaviour, knowledge statements were computed into binary variables to carry out a further binary logistic regression with a multivariate model.

### 2.5. Ethical Considerations

Ethical approval for this study was obtained from the Rowett Institute Ethics Panel and the University of Aberdeen Research Governance (812). All subjects gave informed consent, in line with the Declaration of Helsinki and all researchers had in-date certificates of ethical research training. All data from the survey was completely anonymised, e.g., all postcode data was removed. The data will also be made publicly available on the Open Science Framework, which participants gave their consent to before participation.

## 3. Results

### 3.1. Participant Characteristics

The sample consisted of 1177 adults aged between 18 and 89 years, 57.4% of which were from England and 45.8% were from Scotland. The mean age of the population was 44.0 years (standard deviation (SD) 16.4), with 65.4% of the sample identified as female and 33.8% identified as male. A summary of the participant characteristics is shown in Table 1. Of the total population 47.4% (*n* = 561) were found to purchase PBAF in their food shop. In Scotland only 42.6% purchased PBAF whilst in England 57.4% did. Overall, 88.2% purchased wholefood plant proteins. 9.9% (*n* = 117) and 2.5% (*n* = 30) subjects disclosed they were vegetarian and vegan, which is above the UK average rates of vegetarianism and veganism of 2.9% and 0.4%, respectively, in 2014, for households where the respondent was born between 1930 and 1974 [22]. However a market research portal recently suggested that current rates lie within 7% (vegetarianism) and 4% (veganism) in the UK [23]. 

### 3.2. Purchasing Trends of Plant-Based Foods

#### 3.2.1. Plant-Based Alternative Food Products

When looking at purchasing trends, we have presented the 10 most popular products (Figure 1), with the three most frequently purchased PBAFs being “other meat free burgers and sausages” “mycoprotein burgers and sausages” followed by mycoprotein mince and meatballs. They all depict meat free substitutes for ultra-processed meat products. Four products were mycoprotein products. It can be seen, that more PBAF were purchased in England than in Scotland, which was statistically significant (*p* = 0.041). 

#### 3.2.2. Wholefood Plant-Proteins

The three most frequently purchased wholefood plant proteins were canned baked beans (a traditional food item in the UK), frozen garden peas and canned kidney beans (Figure 2). Amongst the rest of the ten most frequently purchased wholefood proteins were kitchen staples such as frozen petit pois, fresh and frozen green beans, dried lentils, canned chickpeas, black eyed peas and garden peas. Green beans were the only fresh item, whereas half of food items were canned, falling under the processed food category [24]. Scottish people bought significantly more dried lentils (*p* < 0.001) than participants from England. 

### 3.3. Sociodemographic Predictors of Plant-Protein Purchasing

#### 3.3.1. Plant-Based Alternative Food Products

In a fully adjusted multivariate model an independent factor for purchasing PBAF was country of residence (Table 2). English participants were 32% more likely to purchase PBAF (*p* < 0.025). A further independent factor in the fully adjusted model was age. Participants from the Baby Boomer and Traditionalists age group were less likely to purchase PBAF (*p* < 0.001, *p* = 0.027). Moreover, participants with a Black/African or Caribbean background were more likely to purchase PBAF than White participants (OR 3.88, *p* = 0.008). Furthermore, Asian and multiple ethnicities or other were more likely to purchase PBAF than the white population (OR 2.28, *p* = 0.015; OR 2.72, *p* = 0.031).

#### 3.3.2. Wholefood Plant Proteins

When examining the factors regarding the purchasing behaviour of wholefood plant protein products, as depicted in Table 3, the main factor with an overall significant effect in the multivariate model was IMD. Participants with low deprivation (IMD 1) were significantly more likely to purchase wholefood plant-proteins (OR 3.46, *p* = 0.001) than those from high deprivation (IMD 5). Moreover, people from the Baby Boomer generation were less likely to purchase wholefood plant-proteins than those in Generation Z (OR 0.38, *p* = 0.020). However, age did not have an overall effect in the multivariate analysis nor univariate analysis. 

#### 3.3.3. Purchase-Amounts of PBAF and Wholefood Plant Protein Product Types 

When examining the number of foods purchased from both PBAF and wholefood plant protein groups (Table 4), it could be seen that, overall, the population indicated they purchased an average of 2.06 PBAF product types (SD 3.21, median 0.00, interquartile range (IQR) 0, 3) and a mean of 6.92 products (SD 4.57, median 7, IQR 3, 11) of wholefood plant-proteins types in their general grocery shop (out of 18 PBAF and 24 wholefood plant-protein product categories to choose from in the survey). English participants purchased a median of 1 product (IQR 0, 3), whereas the median value for Scottish consumers was 0 (IQR 0, 3) with at statistically significant distribution (*p* = 0.022). Millennials and Generation Z purchased a mean of 2.55 (SD 2.26) and 2.66 (SD 3.70) products (both with a median of 1, IQR 0, 4), whereas Generation X, Baby Boomers and Traditionalists averaged at 2.17 (SD 3.36, median 0, IQR 0, 3), 1.18 (SD 2.43, median 0, IQR 0, 1) and 0.98 (SD 2.14, median 0, IQR 0, 1) products respectively and the difference in this distribution was statistically significant (*p* < 0.001). 

Furthermore, the median purchasing score for wholefood plant-protein types from individuals from low deprivation had a median (IQR) of 8 (5,11) items, whereas individuals from high deprivation had a median of 6 (IQR 3, 11) and overall, the distribution across different IMD groups was found to be significant (*p* = 0.013). This suggests significantly different purchasing patterns between IMD groups. 

### 3.4. Protein Knowledge

Overall, 69.8% of participants recognised lentils as a good source of protein. Also, 88.5% of participants agreed that protein is important for a healthy body and 87% saw protein as important for body muscle. However only 58.5% recognized eating a more plant-based diet as healthy and only 65.8% see eating a plant-based diet as being good for the planet. 

When comparing different sociodemographic factors with agreement to protein knowledge statements (Table 5), it was evident that people from low deprivation areas (IMD 1) are significantly more likely to recognise lentils as a good source of protein (OR 1.94, *p* = 0.003) and more likely to recognise a plant-based diet as being healthy (OR 1.79, *p* = 0.004) than participants from high deprivation areas (IMD 5). A further independent factor affecting protein knowledge in the different multivariate models was gender. Females were more likely to agree that lentils are a good source of protein (OR 1.7, *p* < 0.001), more likely to see a plant-based diet as being good for the planet (OR 1.51, *p* = 0.002) and more likely to recognise a plant-based diet as being healthy (OR 1.59, *p* < 0.001) than men. Finally living in Scotland increased the odds of recognising lentils as good source of protein and participants from England were less likely to recognise lentils as good source of protein (OR 0.75, *p* = 0.028).

## 4. Discussion

### 4.1. PBAF as Ultra-Processed Plant-Based Food

The current study shares novel data from the UK about plant protein purchase behaviours and attitudes, with emphasis on geographical region and SES for PBAF. The self-reported behaviour revealed that purchasing rates for PBAF were relatively high, with 47.4% placing itself between values of a previous UK survey, where 13.1% reported to consume PBAF, from the National Diet and Nutrition survey (NDNS) between 2017–2019 [12]. More recent market research, from 2022, reports that around 65% UK consumers have tried meat-free foods [25]. However, self-reported purchasing does not necessarily translate to actual consumption, and this is a limitation of the current study. When it comes to sustainable food intentions and behaviours, the apparent contradiction between what consumers say and do, has been described as the ‘say-do’ gap [26]. Although intentions are a significant predictor of sustainable behaviour, the solution to this issue is combined data on self-reported attitudes on PBAF, that also link to purchasing behaviour and consumption patterns. Additionally, Culliford and Bradbury [27] and Panzone et al. [28] also suggest that intended sustainable beliefs and shopping baskets do not necessarily match, because consumers are unaware of how their shopping habits are not always in line with actual environmental benefit. When comparing the current results for PBAF items, a recent observational study identified a trend for plant-based burgers, sausages and mince to be the most popular plant-based meat alternatives [29]. This could be because more and more consumers are open to the idea of purchasing meat-free alternatives, that are similar to a meat product, with reasons supporting choice, ranging from health or environmentalism [7]. These PBAF, especially meat substitutes, pose as a large economic opportunity for retailers and producers, with sales revenues consistently rising into the billions [30]. Recently, to promote plant-based eating and increase sales, PBAF products are being more strategically placed (adjacent to, or integrated into, the meat aisles) on UK shop floors. This follows examples from intervention studies that found an increase in sales [31,32]. However, these popular plant-based burgers and sausages can be classed as ultra-processed foods, following the NOVA classification [24,33]. Being an industrially modified food substance with additives [24] these ultra-processed foods bear the risks of being high in salt, fat or sugar. A recent cross-sectional study analysing plant-based meats in the UK, found plant-based meat products to have significantly higher salt levels in five out of six examined categories than their meat-counterpart products, and nearly 75% of plant-based products not achieving national salt reduction target recommendations. On the positive side, they were found to have significantly less saturated fat, total fat and significantly more fibre [34]. In another study in Australia, similar results for a more favourable nutrient profile in terms of fats and fibre were found. However, only 55% of plant-based sausages were found to meet reformulation targets for salt, whereas it was met in 90% of plant-based crumbed/battered meat/poultry products [35]. An online audit in Ireland showed plant-based products to be similar or higher in salt than meat products, but again a source of fibre and lower in saturated fat, total fat and energy [36]. Finally in Sweden similar advantages and disadvantages were shown, of plant-based meat alternatives being higher in fibre and lower in saturated fats, but both meat and plant-based products can contribute highly to intakes of salt within recommended intake levels [37]. 

Products that are labelled as environmentally sustainable may not support healthier food choices. Despite having lower GHG emissions and incorporating ingredients that are favourable in terms of environmental sustainability, there is a need to clearly identify their health benefits and nutritional content [38]. 

### 4.2. Wholefood Plant Protein (Legume) Purchasing and Food Inequality

Substantial evidence exists which highlights the link between food inequality, socio-economic background and poor diet, mostly focused on fruit and vegetable consumption [16]. The evidence base identifies low-socioeconomic background as the “single most consistent risk factor for an unhealthy diet” [17]. The present study also suggests that a link exists between purchasing of wholefood plant proteins, specifically legumes and socio-economic background, when applying IMD score. Food and health inequalities are apparent in the UK, with adults and children from more deprived areas having diets lower in diet quality, with lower intakes of fibre, vegetables, fruit and oily fish [4,18]. This study did not try to quantify the amounts purchased nor consumed across IMD area, and further data is needed to assess intake of legumes in the UK, as well as associated sociodemographic factors including deprivation. To study the issue of deprivation affecting intake of legumes in more detail further research and data would be needed regarding education, employment, income and urban/rural location. We have previously reported intake data, from the UK National Diet and Nutrition Survey (NDNS) program, running 2008–2019, mean (SD) legume intake within the United Kingdom was 26.7 ± 29.6 g/day [39]. Legumes are not the first choice of protein source for people transitioning to a plant-based diet and the current data also support the notion that PBAF that mimic meat products, such as burgers and sausages, require less cooking skills. In their recent study, Alae-Carew et al. report that beans and pulses were found to make up only 0.8% of total daily energy intake when analysing NDNS data from 2017–2019 [12]. Similarly, Lonnie et al. highlight NDNS data from 2013–2014 in their review, that plant-proteins in UK diets are mainly derived from cereal and cereal products (such as bread, pasta, rice) contributing up to 25% to daily protein intakes [40]. The inclusion of more wholefood protein sources such as beans and pulses would not only increase the sustainability of individual diets, but also provide the health benefits of legumes [41], as well as being an affordable wholefood option [42]. Legumes have more fibre than animal derived protein, and are higher in fibre and protein content when compared with commonly consumed cereals and grains [42]. In the UK, the wider population fails to meet recommended intakes of fruit, vegetables and fibre [43,44]. For example, a portion of cooked lentils acts as great source of protein, containing 7.6 g of protein as well as 1.9 g of dietary fibre [45] and can be counted as one of the five-a-day according to the Eatwell guide [46]. However, despite encouraging the consumption of plant-based protein sources, limited guidance is given by the Eatwell guide about recommended intakes [40]. The UK Vegetarian Society has recently published the Vegetarian Eatwell guide [47], to include guidance on beans and pulses for protein. 

### 4.3. Knowledge and Food Choice

Purchase of wholefood plant-based proteins is associated with knowledge. We highlight in the current study that participants from the highest deprived population group recognised lentils (one of the wholefood protein purchasing options in the survey) significantly less as a source of protein, in contrast to participants from an area of low deprivation. However, food choice is affected by a multitude of factors, not only knowledge. Knowledge is amongst the cognitive factors of individual characteristics, next to food-related or societal characteristics that affect food choice [48]. The National Food Strategy reports that knowledge and cooking skills have decreased throughout our society due to the rise of pre-packaged, pre-prepared, convenient food items [4]. We report a significant effect for country, gender, and age regarding knowledge of lentils as source of protein in this study. Future research could not only focus on whether knowledge about plant protein options is present but also whether cooking or culinary skills are available to prepare these foods [40]. Moreover, consumers from areas of low deprivation were significantly more likely to recognise a plant-based diet as being healthy, compared to those from high deprivation areas. For gender, females were significantly more knowledgeable and more likely to recognise plant-based diets as healthy, than males. Lack of knowledge, or more specifically, low awareness of plant-based proteins is a barrier for consumption, highlighted in a recent systematic review [49]. Interestingly, a recent policy report highlighted that general public concerns still remain that a fully plant-based diet could be nutritionally inadequate [5]. Additionally, insufficient knowledge about the environmental sustainability benefits of plant-based diets and poor awareness about environmental impacts of meat-consumption have been repeatedly reported in the wider literature [15,50]. These pose as opportunities for targeted public health education [51]. Public health policy makers need to raise awareness and educate around the nutritional as well as environmental benefits of plant-based wholefoods, targeted for local communities facing food insecurity. This approach to support affordable, healthy, and environmentally sustainable foods will contribute to reducing the existing diet and health inequalities. It is recognised that there is cultural and geographical influence on food choice, and our current data would support future exploration of the reasons why age is a barrier for consumption of plant-based proteins. We have previously highlighted [50] that older age groups report the main obstacles for making plant protein and specifically legume protein their preference, as lack of trust in products, unethical production, poor sensory qualities in terms of product taste, and perceived lack of healthiness. Lower intakes of plant protein have been associated with being male, having a higher income, lower education level and not placing importance on healthy eating [52,53,54]. A rapid transformation to a predominately plant-based diet is unlikely to be feasible on the global scale. However, consumers are becoming increasingly aware of the health benefits of predominantly plant-based diets, which have been associated with lowering the risk of type 2 diabetes, cardiovascular diseases, hypertension, obesity, metabolic syndrome, and all-cause mortality in prospective cohort studies [55,56,57,58,59]. Previous literature also suggests that information alone will not be enough to bring about changes in behavior; population-level sustainable dietary advice or interventions may not produce the same effects in high- and lower-income groups [60]. 

## 5. Conclusions

A limitation of this study is that it is based on a survey asking for self-reported behaviour and knowledge related to diet and health, posing a risk of reporting bias, as well as social desirability bias [61]. Furthermore, the survey could have been completed by someone who purchases plant-based foods for other members of their household, but not necessarily plant-based products for themselves. The nature of the questionnaire as a qualitative survey assessing reported purchasing, did not assess or quantify individual or household consumption. It also did not quantify specific quantities purchased or re-purchasing rates of individual or branded products. 

This study highlights the need for further research in plant-based proteins in order to increase consumption of PBAF, to promote both public and planetary health. Understanding barriers to purchasing wholefood plant proteins will demand more understanding of the associated sociodemographic factors. More research is needed to provide evidence on the effect of deprivation and other sociodemographic variables affecting consumption of plant-based proteins such as legumes. To reduce food and health inequalities, requires sustained behaviour change towards healthier, more sustainable diets. The findings of this study contribute to the expanding evidence base for consumer knowledge and choice around plant-based alternative foods. We highlight the importance of social deprivation on food choice and how this contributes to food-based inequalities. Novel plant-based alternative food products are not necessarily environmentally sustainable and healthy. Due to lack of consumer knowledge, PBAF can be surrounded by a “health-halo” [62] when in fact they are often ultra- processed and high in salt. This poses not only as opportunity for further research but also as an opportunity for policy makers to continue to act and to work on implementing meaningful recommendations for accessible and affordable aspects of plant-based diets, targeted for communities. 

## Figures and Tables

**Figure 1 nutrients-14-04706-f001:**
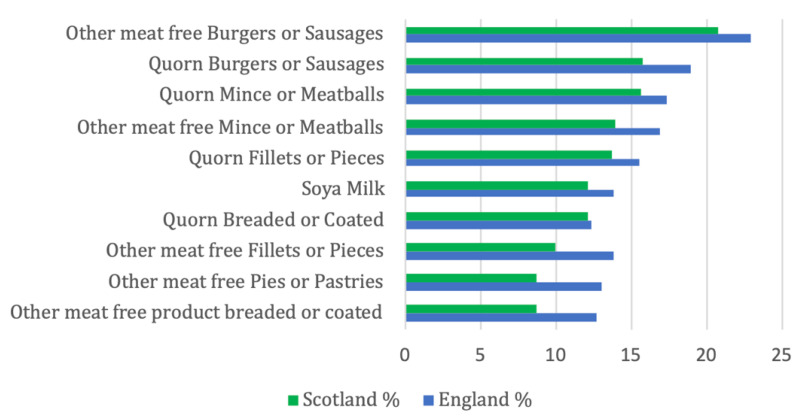
Ten most frequently purchased plant-based alternative food items.

**Figure 2 nutrients-14-04706-f002:**
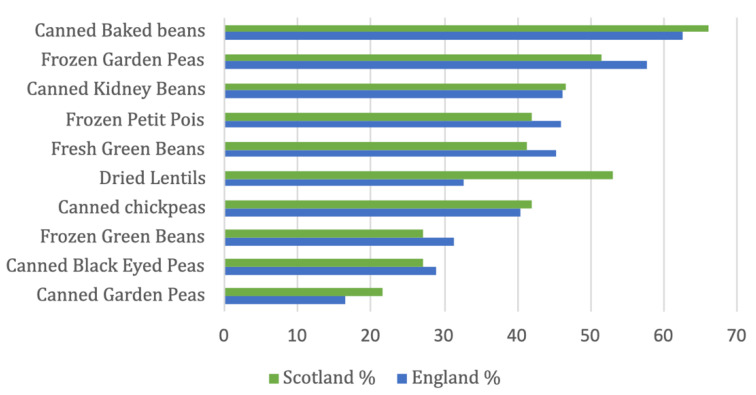
Ten most frequently purchased wholefood plant proteins.

**Table 1 nutrients-14-04706-t001:** Participant characteristics.

		Overall *n* (%)	Consumption of Plant-Based Alternative Foods *n* (%)	Consume Wholefood Plant-Proteins *n* (%)
	All	1177 (100)	561 (47.4)	1038 (88.2)
Country	England Scotland	638 (54.2) 539 (45.8)	322 (57.4) 239 (42.6)	561 (54) 477 (46)
Gender	Male	398 (33.8)	177 (31.6)	356 (34.3)
	Female	770 (65.4)	378 (67.4)	674 (64.9)
	Other *	9 (0.8)	6 (1.1)	8 (0.8)
Age	Generation Z (18–23)	113 (9.6)	63 (11.2)	105 (10.1)
	Millennials (24–39)	406 (34.5)	240 (42.8)	365 (35.2)
	Generation X (40–55)	328 (27.9)	155 (27.6)	286 (27.6)
	Baby Boomers (56–74)	289 (24.6)	90 (16%)	244 (23.5)
	Traditionalists (75+)	41 (3.5)	13 (2.3)	38 (3.7)
Ethnic background	White	1080 (91.8)	490 (87.3)	948 (91.3)
	Black/African/Caribbean	27 (2.3)	22 (3.9)	24 (2.3)
	Asian	43 (3.7)	29 (5.2)	41 (3.9)
	Multiple ethnicities and Other **	27 (2.3)	20 (3.6)	25 (2.4)
Index of Multiple Deprivation level	1 (low deprivation)	192 (16.3)	99 (17.9)	183 (17.8)
	2	199 (16.9)	96 (17.3)	177 (17.2)
	3	244 (20.7)	110 (19.9)	210 (20.4)
	4	269 (22.9)	126 (22.7)	233 (22.7)
	5 (high deprivation)	261 (22.2)	123 (22.2)	224 (21.8)
	Missing	12 (1.0)	12 (1.0)	12 (1.0)

* other refers to: non-binary, prefer not to say. ** other refers to: Arab, prefer not to say, or other.

**Table 2 nutrients-14-04706-t002:** Sociodemographic factors of PBAF purchasing.

		Univariate Analysis Unadjusted Models		Multivariate Analysis Fully Adjusted Model	
Variables		OR (95% CI)	*p* Value	OR (95% CI)	*p* Value
Country	Scotland *				
	England	1.2 (1.01–1.61)	0.036	1.32 (1.04–1.69)	0.025
Overall Effect			0.036		0.025
Gender	Male *				
	Female	1.20 (0.94–1.54)	0.134	0.95 (0.73–1.23)	0.691
	Other **	2.50 (0.62–10.1)	0.200	1.21 (0.28–5.22)	0.800
Overall Effect			0.175		0.880
Age (Generation; years)	Generation Z (18–23) *				
	Millennials (24–39)	1.15 (0.75–1.75)	0.522	1.40 (0.90–2.20)	0.138
	Generation X (40–55)	0.71 (0.46–1.1)	0.120	0.88 (0.56–1.40)	0.599
	Baby Boomers (56–74)	0.36 (0.23–0.56)	<0.001	0.43 (0.27–0.70)	<0.001
	Traditionalists (75+)	0.37 (0.17–0.78)	0.010	0.41 (0.19–0.90)	0.027
Overall Effect			<0.001		<0.001
Ethnic Background	White *				
	Black/African/Caribbean	5.30 (1.99–14.1)	<0.001	3.88 (1.42–10.6)	0.008
	Asian	2.49 (1.30–4.77)	0.006	2.28 (1.17–4.43)	0.015
	Multiple ethnicities and Other ***	3.44 (1.44–8.20)	0.005	2.72 (1.09–6.75)	0.031
Overall Effect			<0.001		<0.001
IMD level	5 (High Deprivation) *				
	1 (Low Deprivation)	1.20 (0.82–1.73)	0.351	1.39 (0.94–2.05)	0.101
	2	1.1 (0.72–1.51)	0.813	1.13 (0.76–1.65)	0.551
	3 (Medium Deprivation)	0.92 (0.65–1.31)	0.645	1.03 (0.71–1.48)	0.884
	4	0.99 (0.70–1.40)	0.947	0.98 (0.69–1.40)	0.908
Overall Effect			0.746		0.424

* First covariate in each sociodemographic group is comparator, therefore there are no results displayed in comparator row. ** non-binary, prefer not to say. *** Arab, prefer not to say, or other. PBAF, plant-based alternative foods; IMD, index of deprivation; OR, odds ratio; CI, confidence interval.

**Table 3 nutrients-14-04706-t003:** Sociodemographic factors of wholefood plant-protein purchasing.

		Univariate Analysis Unadjusted Models		Multivariate Analysis Fully Adjusted Model	
Variables		OR (95% CI)	*p* Value	OR (95% CI)	*p* Value
Country	Scotland *				
	England	0.947 (0.66–1.35)	0.764	0.96 (0.66–1.38)	0.809
Overall Effect			0.764		0.809
Gender	Male *				
	Female	0.83 (0.56–1.22)	0.337	0.72 (0.48–1.07)	0.106
	Other **	0.94 (0.12–7.73)	0.957	0.60 (0.07–5.25)	0.640
Overall Effect			0.629		0.263
Age (Generation; years)	Generation Z (18–23) *				
	Millennials (24–39)	0.68 (0.31–1.50)	0.334	0.69 (0.31–1.55)	0.369
	Generation X (40–55)	0.52 (0.24–1.14)	0.103	0.52 (0.23–1.17)	0.115
	Baby Boomers (56–74)	0.41 (0.19–0.91)	0.028	0.38 (0.17–0.86)	0.020
	Traditionalists (75+)	0.97 (0.24–3.83)	0.96	0.84 (0.21–3.40)	0.803
Overall Effect			0.078		0.052
Ethnic Background	White *				
	Black/African/Caribbean	1.11 (0.33–3.75)	0.862	0.97 (0.28–3.37)	0.957
	Asian	2.85 (0.68–11.9)	0.151	2.47 (0.58–10.4)	0.221
	Multiple Ethnicities and Other ***	1.74 (0.41–7.43)	0.454	1.31 (0.29–5.94)	0.722
Overall Effect			0.456		0.655
IMD level	5 (High Deprivation) *				
	1 (Low Deprivation)	3.36 (1.58–7.14)	0.002	3.46 (1.62–7.40)	0.001
	2	1.33 (0.76–2.33)	0.322	1.29 (0.73–2.29)	0.379
	3 (Medium Deprivation)	1.02 (0.62–1.69)	0.938	1.05 (0.63–1.74)	0.852
	4	1.10 (0.65–1.75)	0.791	1.05 (0.63–1.72)	0.862
Overall Effect			0.022		0.021

* First covariate in each sociodemographic group is comparator, therefore there are no results displayed in comparator row. ** other refers to: non-binary, prefer not to say. *** other refers to: Arab, prefer not to say, other.

**Table 4 nutrients-14-04706-t004:** Distribution of Purchase Amounts of product types within sociodemographic groups.

Sociodemographic Groups	Cumulative PBAF Score			Cumulative Wholefood Plant-Protein Score		
	Mean Rank	Median (IQR)	Mean (SD)	Mean Rank	Median (IQR)	Mean (SD)
Country						
England	608.27	1.00 (0, 3)	2.27 (3.43)	602.22	7.00 (3.75, 11)	7.13 (4.66)
Scotland	566.19	0.00 (0, 3)	1.82 (2.91)	573.35	6.00 (3, 11)	6.68 (4.45)
*p* value (Mann–Whitney test)	0.022			0.145		
Gender						
Male	570.56	0.00 (0, 3)	1.85 (2.91)	566.99	6.00 (3, 11)	6.63 (4.47)
Female	597.14	0.00 (0, 3)	2.15 (3.31)	599.87	7.00 (3, 11)	7.06 (4.60)
Other *	708.39	1.00 (0, 8)	3.89 (5.58)	632.44	8.00 (3.5, 13)	8.11 (5.95)
*p* value (Kruskal–Wallis test)	0.203			0.270		
Age						
Generation Z (18–23)	645.93	1.00 (0, 4)	2.66 (3.70)	563.29	6.00 (3, 11)	6.50 (4.10)
Millennials (24–39)	657.36	1.00 (0, 4)	2.55 (2.26)	639.47	7.00 (5, 11)	7.62 (4.68)
Generation X (40–55)	591.41	0.00 (0, 3)	2.17 (3.36)	580.86	6.50 (3, 11)	6.82 (4.60)
Baby Boomers (56–74)	484.34	0.00 (0, 1)	1.18 (2.43)	545.76	6.00 (2.5, 10)	6.33 (4.57)
Traditionalists (75+)	473.62	0.00 (0, 1)	0.98 (2.14)	530.01	6.00 (3, 9)	6.12 (3.64)
*p* value (Kruskal–Wallis test)	<0.001			0.003		
Ethnic Background						
White	575.90	0.00 (0, 3)	1.97 (3.15)	581.69	7.00 (3, 11)	6.83 (4.56)
Black/African/Caribbean	775.81	3.00 (1, 4)	3.48 (4.25)	623.65	7.00 (3, 12)	7.37 (4.85)
Asian	710.97	2.00 (0, 5)	3.00 (3.58)	716.22	10.00 (6, 12)	8.56 (4.44)
Multiple Ethnicities and Other **	731.93	2.00 (0, 4)	2.89 (3.23)	644.11	6.00 (4, 11)	7.63 (4.55)
*p* value (Kruskal–Wallis test)	<0.001			0.056		
IMD						
1 (Low Deprivation)	600.09	1 (0, 3)	2.07 (2.95)	657.89	8.00 (5, 11)	8.01 (4.45)
2	592.82	0 (0, 3)	2.25 (3.44)	576.94	6.00 (3, 11)	6.78 (4.33)
3 (Medium Deprivation)	566.54	0 (0, 3)	2.02 (3.47)	578.09	7.00 (3, 11)	6.87 (4.66)
4	585.20	0 (0, 3)	2.07 (3.08)	570.97	7.00 (3, 11)	6.79 (4.72)
5 (High Deprivation)	576.07	0 (0, 3)	1.96 (3.14)	549.52	6.00 (3, 11)	6.42 (4.50)
*p* value (Kruskal–Wallis test)	0.808			0.013		

* other refers to: non-binary, prefer not to say. ** other refers to: Arab, prefer not to say, or other. IQR, interquartile range; SD, standard deviation.

**Table 5 nutrients-14-04706-t005:** Sociodemographic groupings showing agreement with statements about protein-multivariate model analysis.

		Lentils Are a Good Source of Protein		Protein Is Important for a Healthy Body		Protein Is Important for Body Muscle		Eating a Plant-Based Diet Is Good for the Planet		Eating a More Plant-Based Diet Is Healthy	
Variables		OR (95% CI)	*p* Value	OR (95% CI)	*p* Value	OR (95% CI)	*p* Value	OR (95% CI)	*p* Value	OR (95% CI)	*p* Value
Country	Scotland *										
	England	0.75 (0.57–0.97)	0.028	0.69 (0.47–1.01)	0.056	0.87 (0.61–1.24)	0.439	0.98 (0.76–1.26)	0.873	1.18 (0.92–1.50)	0.194
	Overall Effect		0.028		0.056		0.439		0.873		0.194
Gender	Male										
	Female	1.70 (1.29–2.23)	<0.001	1.20 (0.81–1.78)	0.368	1.01 (0.70–1.48)	0.946	1.51 (1.16–1.96)	0.002	1.59 (1.23–2.05)	<0.001
	Other **	1.50 (0.34–6.72)	0.593	1.52 (0.18–13.0)	0.703	0.51 (0.10–2.66)	0.425	1.52 (0.36–6.50)	0.570	2.06 (0.48–8.96)	0.334
	Overall Effect		<0.001		0.645		0.711		0.009		0.002
Age (Generation; years)	Generation Z (18–23) *										
	Millennials (24–39)	2.09 (1.32–3.31)	0.002	1.35 (0.74–2.48)	0.332	1.00 (0.53–1.89)	1.000	1.35 (0.86–2.13)	0.198	1.04 (0.66–1.62)	0.873
	Generation X (40–55)	1.94 (1.21–3.12)	0.006	1.24 (0.66–2.33)	0.497	1.12 (0.58–2.15)	0.764	1.44 (0.89–2.30)	0.134	1.18 (074–1.88)	0.485
	Baby Boomers (56–74)	1.72 (1.06–2.80)	0.027	1.76 (0.89–3.47)	0.102	1.00 (0.51–1.96)	0.998	0.85 (0.53–1.36)	0.493	0.78 (0.49–1.25)	0.303
	Traditionalists (75+)	2.52 (1.09–5.81)	0.031	3.45 (0.75–15.9)	0.112	1.37 (0.42–4.49)	0.604	1.35 (0.62–2.95)	0.454	0.99 (0.47–2.10)	0.981
	Overall Effect		0.025		0.326		0.970		0.016		0.180
Ethnic background	White *										
	Black/African/Caribbean	3.35 (1.10–10.2)	0.033	0.56 (0.21–1.50)	0.249	1.34 (0.39–4.65)	0.643	1.14 (0.48–2.73)	0.762	1.19 (0.51–2.76)	0.690
	Asian	1.69 (0.80–3.56)	0.168	0.71 (0.30–1.66)	0.427	1.12 (0.43–2.92)	0.818	1.54 (0.76–3.13)	0.236	0.72 (0.39–1.35)	0.310
	Multiple Ethnicities and Other ***	0.71 (0.30–1.65)	0.419	0.52 (0.18–1.46)	0.213	0.84 (0.27–2.57)	0.757	0.55 (0.25–1.24)	0.149	0.34 (0.15–0.77)	0.010
	Overall Effect		0.068		0.377		0.945		0.290		0.051
IMD	5 (High Deprivation) *										
	1 (Low Deprivation)	1.94 (1.26–2.99)	0.003	2.01 (1.06–3.80)	0.032	1.77 (0.99–3.18)	0.055	1.45 (0.97–2.17)	0.070	1.79 (1.21–2.65)	0.004
	2	1.46 (0.97–2.20)	0.073	1.36 (0.78–2.40)	0.282	1.17 (0.70–1.98)	0.549	1.22 (0.82–1.80)	0.323	1.22 (0.83–1.78)	0.309
	3 (Medium Deprivation)	1.46 (0.99–2.15)	0.055	1.67 (0.96–2.90)	0.069	1.37 (0.82–2.27)	0.229	1.49 (1.03–2.18)	0.036	1.48 (1.03–2.13)	0.032
	4	1.11 (0.77–1.60)	0.595	1.28 (0.77–2.10)	0.342	1.44 (0.87–2.37)	0.152	1.17 (0.81–1.67)	0.402	1.37 (0.97–1.95)	0.077
	Overall Effect		0.020		0.197		0.338		0.225		0.044

* First covariate in each sociodemographic group is comparator, therefore there are no results displayed in comparator row. ** other refers to: non-binary, prefer not to say. *** other refers to: Arab, prefer not to say, or other.

## Data Availability

Contact Alex Johnstone for access to anonymised data.

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
