# Peer review of "Self-Reported Purchasing Behaviour, Sociodemographic Predictors of Plant-Based Protein Purchasing and Knowledge about Protein in Scotland and England"

_nutrients, 2022, doi:10.3390/nu14214706_

Round 1

Reviewer 1 Report

Thank you very much for the review invitation for the paper (Nutrients Plant Protein Knowledge and Socio-Economic Drivers of Purchase Behaviours for Consumers in Scotland and England). I have carefully and thoroughly read the related files. The paper focused on purchase behaviours of plant based foods, it is beneficial for consumers to choice suitable plant-based foods. But there are still some suggestions. The suggestions are as follows:

1) There is no analysis in the paper on the reasons why consumers of different age groups (Gen Z, Millennials, etc.) buy various products, whether they may be tasty, nutritious and healthy, recommended by others or other reasons? For the section4.1 only the purchase rate is analysed, has the repurchase rate been investigated?

2) How did the test persons obtain the online questionnaire? If it was shared with specific people (e.g. food manufacturers or construction site workers) who themselves have different knowledge of plant protein foods, how was the randomness of the questionnaire ensured?

3) For the section 3.2.1, the products involved, whether there are other brands,; e.g. Beyond Meat Burger, Impossible Burger, etc.. Some plant-based seafoods (e.g. plant-based salmon etc.) are also produced, which are not covered in the study. Why is it most acceptable for burger sausages, because of dietary habits or for other reasons?

4) There are many different types of commercially available products, have researchers measured their nutritional content , e.g. protein/fat/carbohydrate content etc. ?  Whether they are compared with relevant international standards or WHO/FAO recommended data. And is it reasonable to conclude directly that they are high in salt, fat and carbohydrates? It is well known that plant-based foods/meat simulants/meat substitutes are green & sustainable foods, mainly due to the advantages of high proteins, low fats and low calories content, as well as the addition of many vitamins and mineral elements, although they are ultra-processed foods, they have many nutritional elements. In the

5) How are poverty levels classified, based on income or other factors? There is no doubt that there are differences in economic levels and purchasing power will definitely vary. Then the price of commercially available PABF products needs to be added and further analysed.

6) The section 4.3 mentions that many people are not sufficiently knowledgeable about plant protein foods, how can this be changed or improved? Some specific measures and suggestions should be suggested, and whether any relevant scholars have made similar observations or suggestions.

Author Response

AR-Thank you for your review and positive comments. All queries are appreciated to improve the paper and answered to below the individual comments. We have noted these with ‘AR’ as Author Response.

Reviewer 2 Report

The paper offers some very interesting insights into purchase behavior towards plant-based food and plant-based proteins. The lack of the literature in this field and the importance of food-related topics provide extra significance to this study. However, several ambiguities arose while reading the paper. Please, see the comments.

Originality: The originality of the study is highlighted in the appropriate manner.

Introduction and literature review: The introduction section provides needed information regarding the importance of the research, research gap and main aims of the study. However, it lacks proper literature review. It is necessary to cite more sources dealing with this topic, in order to provide better insight into main conclusions of previous studies, especially those investigating the main factors that shape consumers’ behavior toward plant-based diets. This could be achieved by placing some parts of the discussion in this section. Still, more research analyzing the influence of factors that affect plant-based food purchase should be cited.

Methodology: The methodology used the research is described in quite fair manner. However, the data analysis is predominantly based on descriptive statistics and multivariate analysis. The study should be strengthen by implementing another econometric tool that would analyze the strength of impact that various socioeconomic factors (included in this research) have on purchase of plant-based food (e.g. SEM model or other appropriate tool). Besides of that, the methodology section does not explain what sampling technique was used and why it can be considered suitable for this type of research. Please, provide missing information.

Results and Discussion: Some results of the research, provided in tables, are not properly explained in the following text. For example, the authors stated that “people from low deprivation areas 234 are significantly more likely to recognize lentils as a good source of protein (OR 1.94, p =235 0.003) and more likely to recognize a plant-based diet as being healthy (OR 1.79, p = 0.044)”. Less likely compared to which group of consumers? When comparing the likelihood of purchasing certain type of products, it should be clearly specified which groups of consumers are being compared. 

The results are fairly discussed. However, the limitations of the study and future research directions should be placed as part of the concluding section (not discussion).

Practical implications: The theoretical and practical implication of the study are pointed out in a quite fair way.

Quality of communication: The language used in the manuscript is concise and clear in all parts of the paper, except for section results (please see the previous comments). However, there are several more suggestions regarding this issue:

o   The presentation of main findings given in abstract is unnecessarily weighted by adding OR and p values. The abstract should contain 2-3 short and clear sentences that present main findings of the research, without adding information about value of individual parameters.

o   “Plant-based” should be deleted from the key words, as there are already given “plant-based foods” and “plant-based protein”.

o   It is unnecessary to mention the name of the specific brand (QuornTM) as the paper does not provide case study based on the concrete brand, but empirical study of consumers’ behavior and attitudes.

Author Response

AR- We thank the reviewer for their positive comments and opportunity to clarify these points. All queries are appreciated to improve the paper and answered to below the individual comments. We have noted these with ‘AR’ as Author Response.

Round 2

Reviewer 1 Report

Yes, it has been improved

Author Response

Reviewer 1 did not have any further queries. We thank you for taking the time to review our paper. 

Reviewer 2 Report

The authors made certain improvements of the manuscript, which contributed to its quality. However, simple methodology used to analyze data still represents its main disadvantage. Please, see the following comments.

Originality: The originality of the study is highlighted in the appropriate manner.

Introduction and literature review: The introduction section provides needed information regarding the importance of the research, research gap and main aims of the study. However, it lacked proper literature review. The authors added some new references. Although these references are not added in the literature review section (where they should be), but in discussion section, the manuscript now covers appropriate range of literature sources.

Methodology: The methodology used the research is described in quite fair manner. The authors added missing information about sampling technique used in the survey. However, the data analysis is predominantly based on descriptive statistics and multivariate analysis, without using any advanced statistical or econometric tool suitable for this type of research, which represents main deficiency of the manuscript.

Results and discussion: The results are presented in an understandable manner now and properly discussed.

Practical implications: The theoretical and practical implication of the study are pointed out in a quite fair way.  

Quality of communication: The language used in the manuscript is concise and clear. Technical errors are corrected now.

Author Response

Reviewer 2 has commented on the 'Methodology' to suggest advanced statistical approached. While we agree this would be nice, we simply don't have resources to facilitate this approach - sorry ! However, we will take this into account for future survey work, and for future publication in Nutrients. This is a growing field of research. Thank you for your time to improve this paper.